# Progress on a Novel, 3D-Printable Heart Valve Prosthesis

**DOI:** 10.3390/polym15224413

**Published:** 2023-11-15

**Authors:** Filip Schröter, Ralf-Uwe Kühnel, Martin Hartrumpf, Roya Ostovar, Johannes Maximilian Albes

**Affiliations:** 1Department of Cardiovascular Surgery, Heart Center Brandenburg, Brandenburg Medical School Theodor Fontane, 14770 Brandenburg an der Havel, Germany; ralf-uwe.kuehnel@immanuelalbertinen.de (R.-U.K.); martin.hartrumpf@immanuelalbertinen.de (M.H.); roya.ostovar@immanuelalbertinen.de (R.O.); johannes.albes@immanuelalbertinen.de (J.M.A.); 2Faculty of Health Sciences Brandenburg, 14476 Potsdam, Germany

**Keywords:** polymeric heart valves, 3D printing, aortic valve prosthesis

## Abstract

(1) Background: Polymeric heart valves are prostheses constructed out of flexible, synthetic materials to combine the advantageous hemodynamics of biological valves with the longevity of mechanical valves. This idea from the early days of heart valve prosthetics has experienced a renaissance in recent years due to advances in polymer science. Here, we present progress on a novel, 3D-printable aortic valve prosthesis, the TIPI valve, removing the foldable metal leaflet restrictor structure in its center. Our aim is to create a competitive alternative to current valve prostheses made from flexible polymers. (2) Methods: Three-dimensional (3D) prototypes were designed and subsequently printed in silicone. Hemodynamic performance was measured with an HKP 2.0 hemodynamic testing device using an aortic valve bioprosthesis (BP), a mechanical prosthesis (MP), and the previously published prototype (TIPI 2.2) as benchmarks. (3) Results: The latest prototype (TIPI 3.4) showed improved performance in terms of regurgitation fraction (TIPI 3.4: 15.2 ± 3.7%, TIPI 2.2: 36.6 ± 5.0%, BP: 8.8 ± 0.3%, MP: 13.2 ± 0.7%), systolic pressure gradient (TIPI 3.4: 11.0 ± 2.7 mmHg, TIPI 2.2: 12.8 ± 2.2 mmHg, BP: 8.2 ± 0.9 mmHg, MP: 10.5 ± 0.6 mmHg), and effective orifice area (EOA, TIPI 3.4: 1.39 cm^2^, TIPI 2.2: 1.28 cm^2^, BP: 1.58 cm^2^, MP: 1.38 cm^2^), which was equivalent to currently used aortic valve prostheses. (4) Conclusions: Removal of the central restrictor structure alleviated previous concerns about its potential thrombogenicity and significantly increased the area of unobstructed opening. The prototypes showed unidirectional leaflet movement and very promising performance characteristics within our testing setup. The resulting simplicity of the shape compared to other approaches for polymeric heart valves could be suitable not only for 3D printing, but also for fast and easy mass production using molds and modern, highly biocompatible polymers.

## 1. Introduction

Amongst the first types of artificial heart valve replacements were polymeric heart valves [1,2], which are made of flexible, synthetic materials. It was hoped that these would combine the high lifespan of solid mechanical valves with the superior hemocompatibility of flexible biological valves. The recipients of some of these valves survived for more than 25 years without the need for a replacement [3,4]. However, further developments led to the widespread adoption of two competing types of heart valve prostheses: biological valves made from flexible porcine heart valves or bovine pericardium, and mechanical valves made from solid synthetic materials like pyrolytic carbon. Advances in the field of polymer science have recently led to the development of various highly biocompatible and calcification-resistant materials [5,6,7,8], which have enabled a renaissance of the concept, with various groups working on such types, predominantly mimicking the shape of the natural valve or common biological prostheses [5,6,9,10,11,12,13].

Although flexible polymers can mimic the movement of biological structures to some extent, it should not be forgotten that the main focus of polymer research on this topic will likely be the biocompatibility of the material. Since the structural and mechanical probabilities of these materials might somewhat differ from those of natural valve tissue, it is not self-evident that the optimal shape of a polymeric heart valve prosthesis will necessarily be the same as evolution intended for living tissue. The aim of our ongoing research is therefore to develop novel forms of polymeric heart valves that not only mimic the shape of the natural aortic valve but still provide competitive performance characteristics. Based on this line of thought, we explored a completely different development direction for an aortic valve prosthesis called the PIZZA and TIPI valve that we published previously [11]. This type of prosthesis is based on a flat (PIZZA) or conical (TIPI) arrangement of triangular leaflets. These leaflets are fixed on a foldable metal wire frame including inward-facing loops as restrictors that limit the leaflets’ movement to facilitate a unidirectional fluid movement. The leaflets were initially cut from silicone sheets and partly overlapped to reduce leakage. In performance tests, these earlier prototypes showed promising initial results, but they did not reach the values measured for currently used aortic valve bioprostheses (BPs) or mechanical prostheses (MP), as they had a regurgitation fraction of 36.6 ± 5.0% (BP: 8.8 ± 0.3%, MP: 13.2 ± 0.7%) and a systolic pressure gradient of 12.8 ± 2.2 mmHg (BP: 8.2 ± 0.9 mmHg, MP: 10.5 ± 0.6 mmHg; see Table 1).

In addition, some criticism remained regarding the limited reproducibility caused by the manufacturing process, in which silicone sheets are cut and glued onto hand-folded metal wire structures, as well as regarding the presence of the potentially thrombogenic restrictors within the orifice area.

To address these problems, we modified the shape of the prototype in order to ensure unidirectional opening without the need for metal restrictors. Furthermore, we switched our manufacturing process to silicone 3D printing, which allowed us to dramatically increase the precision and reproducibility of our prototype development, as demonstrated previously with a different valve prototype developed by our group [14]. There are a variety of techniques for 3D printing, most of which use rigid polymers. In contrast, there are also some printers available that can use flexible materials such as silicone.

Another aspect worth mentioning is 4D printing, which describes methods for the 3D printing of objects that are able to change their form or physical properties in response to certain stimuli, including but not limited to heat, magnetic fields, and light [15,16,17,18]. In terms of developing polymeric heart valves, this could be of great interest for the creation of stents and similar solid, expandable structures that unfold the valves after delivery through minimally invasive or catheter-based techniques.

In some cases, this has also been achieved with flexible materials such as hydrogels [18,19,20] or blends of polyvinyl chloride and polycaprolactone [21]. Theoretically, it would be interesting to explore the possibilities of such materials for the manufacture of heart valve prostheses, especially if a material could be developed that was able to change its stiffness or shape in response to the shear rates and flow direction of the passing fluid on a timescale in the range of single heartbeats. To the best of our knowledge, however, such a material is not yet available. With the focus of our current research, the possibilities of 4D printing would be predominantly interesting with regard to the minimally invasive delivery of a potential prosthesis. Furthermore, such materials would have to meet mechanical and biocompatibility requirements, in addition to these shape-shifting or shape-memory properties. Therefore, they do not yet play a role in our considerations for the further development of this polymeric valve prosthesis, but that may change in the future.

## 2. Materials and Methods

For our current work, we switched to using a high-resolution Keyence Agilista 3200 W 3D printer (Keyence, Osaka, Japan), which allows for the utilization of silicones with Shore hardness of 35 and 65. The required .stl files were created with FreeCad 0.18. The general development milestones are illustrated in Figure 1. The Keyence Agilista 3200 W uses inkjet technology, which is why a fixed standard set of printing parameters was used for the silicones mentioned, including a layer thickness of 30 µm and a resolution of 635 × 400 dpi. As the 3D files represented solid models, the prints were equivalent to a 100% infill on an FDM printer. The prototypes were created from the selected silicone embedded in a soft, waxy, water-soluble support material. Most of this support material can be physically removed afterwards before the remaining residue is dissolved in water. This means that no solid supports have to be removed from the silicone prototype itself, as is common with other 3D printing techniques. The print quality and reproducibility were checked by visual and haptic inspection, as well as measurement using a caliper gauge where appropriate.

In a first step, the conical TIPI valve was reproduced as a 3D model without overlapping leaflets (TIPI 3.0), as the significantly higher precision of the printing approach compared to the cutting and fixation of silicone sheets opened up new possibilities to enable a tight closure without hindering the leaflet movement by overlapping. A flat disk was added to the outer border as an adapter for the testing device. This structure was then attached to the metal wire restrictor structure inherent to the initial patent using a silicone adhesive.

From this basic prototype, various design modifications were investigated, with the aim of removing the need for the wire restrictor structure. Increased side skirts were added to each triangular leaflet to improve coaptation between them (TIPI 3.1). The thickness of the leaflets was then increased towards their combined base, and a series of incisions were added within the leaflet to support their unidirectional movement (TIPI 3.2). Later, this was reduced to a single incision, leading to the first prototype that ensured unidirectional movement (TIPI 3.3). Finally, the performance was further improved by shifting the cutting direction (TIPI 3.4). The prototypes were produced from silicones with two Shore hardness values (35 and 65), but as the more flexible material (Shore 35) expressed better performance, only the results of these valves are presented here. We also tested increasing the number of triangular leaflets to 8 or 10, which led to inferior results.

To evaluate the valves’ performance, we used an HKP 2.0 hemodynamic testing device (LB Engineering GbR, Berlin, Germany) based on the device presented by Schichl and Affeld [22], which was described in previous publications [11,14,23,24,25]. Our system enables the recording of closing time, closing volume, leakage, cardiac output, regurgitation fraction, and mean systolic pressure gradient (Δp_systolic_). As a benchmark for the ongoing optimization process, we compared the new prototypes with the TIPI 2.2 valve reported previously [11], as well as with an aortic bioprosthesis (Edwards Perimount 23 mm, Edwards Life Sciences, Irvine, CA, USA) and a mechanical prosthesis (23 mm bileaflet valve, St. Jude Medical, Saint Paul, MN, USA) of similar size. The simulation of the aortic pressure and flow profile was performed at a heart rate of 70 bpm and a stroke volume of 70 mL, resulting in a cardiac output of 4.9 L/min. Each valve was measured at least 10 times to ensure the reproducibility of the results. A Phantom Miro C320 high-speed camera (Vision Research, Wayne, IL, USA) installed above the valve was used to record the valve’s movement. This enabled precise observation of the opening and closing process in order to identify weak points and opportunities for improvement.

The resulting data were statistically analyzed with R (R Core Team, Vienna, Austria, 2017) [26]. As the data were numerical, we tested them for normal distribution using the Shapiro–Wilk test, before comparing them using Student’s *t*-test or the Mann–Whitney U test.

We calculated the effective orifice areas according to ISO 5840-2:2021 [27] and Formula (1):(1)$$EOA=\frac{q_{vRMS}}{51.6\times \sqrt{\frac{\Delta p}{\rho }}}$$
where *q_vRMS_* is the root-mean-squared forward flow during the positive differential period, Δ*p* is the mean systolic pressure gradient, and *ρ* is the testing fluid density, which we approximated as 1 g/cm^3^.

## 3. Results

A representative result of the 3D printing after removing the waxy support material is illustrated in Figure 2a. The flat disk surrounding the functional valve itself was added for practical reasons due to the shape of the valve adapter in the hemodynamic testing setup. Figure 2b,c show snapshots of the valve in open and closed states, respectively, during a simulated heartbeat cycle in our hemodynamic testing device. The extent of opening shown in Figure 2b was reached at the peak of the simulated systole. Figure 2c demonstrates the tight closure between the leaflets and the high form stability of the prototype even under pressure during the diastole. A high-speed video of the presented prototype during a simulated cardiac cycle within the hemodynamic testing setup can be accessed at https://data.mendeley.com/datasets/6r8f26t6fp/1 (accessed on 17 August 2023).

The performance results of the working prototypes and the three benchmark valves are presented in Table 1 and Figure 3. It should be noted that TIPI 3.0, which still possessed the metal restrictor structure but lacked the overlapping leaflets of the prototype presented in our previous publication [11], was excluded due to severe leakage caused by the insufficient coaptation area between the leaflets. For the same reason, TIPI 3.2, which was the first model without restrictors, was omitted because of its tendency for a leaflet to flip over, leading to an undesired, bidirectional leaflet movement and backflow of the test medium.

TIPI 3.1, which still included a metal restrictor structure but already included a an enlarged vertical coaptation zone at the edges of each leaflet, showed a pronounced improvement in its regurgitation fraction, demonstrating the success of this design choice in improving the tightness of closure compared to the previously published design (TIPI 2.2). However, the EOA and systolic pressure gradient devolved in an undesired direction and remained significantly different from the target benchmarks (all *p* < 0.001), which we attributed to the remaining obstruction caused by the foldable wire structure.

The first functional prototype without restrictors (TIPI 3.3) mostly eliminated this problem, with systolic pressure gradients (BP: *p* = 0.001, MP: *p* = 0.369) and an EOA comparable to those of mechanical prostheses (BP: *p* = 0.001, MP: *p* = 0.413), while the improved regurgitation fraction was mostly maintained.

The final prototype (TIPI 3.4) improved the regurgitation fraction even more, reaching the performance of the mechanical prosthesis (BP: *p* < 0.001, MP: *p* = 0.063) while retaining the EOA and systolic pressure gradient.

As demonstrated, we were able to drastically improve the performance of our prototype compared to previous designs while approaching the values of commonly used prostheses (BPs and MPs) within our hemodynamic testing setup. While TIPI 3.4 still had a significantly higher regurgitation fraction, a systolic pressure gradient, and a lower EOA compared to the BP, the differences were small and it did not differ from the MP in any of these parameters. Since MPs have been successfully used as heart valve prostheses for decades, the prospects for an in vivo application of the presented valve design are promising.

## 4. Discussion

By modifying our prototype, we were able to create a purely polymeric heart valve prosthesis with in vitro performance characteristics similar to those of currently used aortic valve prostheses. The transvalvular gradients of the TIPI valve were already reasonable in the earlier design stages compared to conventionally stented bioprostheses. However, with the evolutionary steps in the design, we were able to reduce the initially quite disappointing regurgitation fraction markedly, to a degree entirely comparable to existing bioprostheses of comparable sizes.

We do not dispute that mimicking the shape of natural heart valves or proven biological valve prostheses is a logical choice when designing polymeric valves and will lead to excellent results. However, it should be noted that the main focus in the development of polymers for heart valve prostheses is on a combination of biocompatibility, calcification resistance, mechanical properties, and manufacturing concerns [6,9,10], and not on reproducing the mechanical properties of natural tissue. Furthermore, even an advanced polymeric material will not have the regenerative capabilities of living heart valve tissue. We therefore argue that competitive or even optimal designs for valve prostheses made from these polymers could differ significantly from the form that has evolved as the optimal solution for living tissue, making research such as that presented here worthwhile.

The manufacturing price per 3D-printed piece was around EUR 20.90, which is several orders of magnitude cheaper than commercially used valve prostheses. As this prototype has a rather simple shape, it could be manufactured using a mold rather than 3D printing for fast and cheap production. This would also enable the usage of a variety of suitable, flexible, highly biocompatible polymers, including but not limited to polyvinyl alcohol [9,28], polydimethylsiloxane [7], and POSS-PCU [29]. Sheets created from some of these polymers have already been used in in vivo studies in sheep, with promising results, including but not limited to their calcification resistance and the host’s inflammatory response [5,30]. Although these materials might be more expensive than silicone, we are optimistic that the cost savings from large-scale production could still ensure a competitive production price. This could be of particular relevance regarding the creation of a functioning valve prosthesis to be made widely available to the large collective of patients in third-world countries, who do not have access to life-saving valve replacements due to economic constraints.

The implied versatility in terms of the material used in the future is especially relevant because the silicone that we used for this study has well-known limitations for use as valve prosthesis material [6,31,32]. Nonetheless, it was well suited for the present stage of development, as silicones, to the best of our knowledge, are the only flexible materials currently used for commercial, precision 3D printers. They can therefore be used to explore new shapes and concepts relatively quickly before switching to more biocompatible polymers. In another approach, it may be worth investigating other flexible materials that have been used in 3D or 4D printing [19,20,21] for their mechanical properties, biocompatibility, and calcification resistance, so as to potentially identify a material suitable for the 3D printing of prototypes for in vivo testing.

The hemodynamic testing setup has its limitations, especially with regard to long-term reliability studies and the fact that it does not correspond to the rather flexible surroundings in which a valve prosthesis will find itself in vivo. Therefore, the next steps towards clinical development of this valve should include long-term testing as well as studies in a suitable animal model, such as sheep. The latter could also be used to assess whether the presented shape, including the increased coaptation area between the triangular leaflets and their central meeting point, as well as the opening and closing incisions at the leaflet base, pose a risk of hemolysis.

While our current testing device is programmed to simulate the pressure and flow profile at the aortic valve position, future research could include setups that simulate other valve positions. In this way, it could be tested whether the aortic valve prosthesis prototype presented here could also be an option for the mitral, pulmonary, and tricuspid valve positions. Furthermore, we used distilled water at room temperature as a test fluid, which has a similar density (ρ = 1 g/cm^3^) but a different viscosity (η ~ 0.9 to 1.00 mPas at 20–25 °C) compared to blood (ρ ~ 1.05 g/cm^3^ [33], η ~ 3–5.5 mPas at 37 °C and shear rates normal within the heart [34,35,36]). Blood is also a non-Newtonian fluid with shear-thinning characteristics, changing its viscosity in response to shear stress [35,36], and some parameters measured in hemodynamic testing setups have been reported to be affected by these differences [37]. However, since the shear rates at the aortic valve are rather high (>200 s^−1^) [36] and blood loses most of its non-Newtonian behavior above 100–200 s^−1^ [35], this physical property seems to be less of an issue in hemodynamic tests simulating the aortic valve’s position.

Nevertheless, it might be appropriate to repeat the experiments with a fluid of higher viscosity—for example, a glycerol solution or a suitable shear-thinning substitute, e.g., solutions containing xanthan or polyacrylamide [38,39,40].

The fully polymeric and flexible nature of the valve presented here could also facilitate transcatheter or at least minimally invasive implantation, although the relative thickness of the leaflets somewhat limits the extent to which it can be compressed. A stent or similar structure to hold such a valve in place would also have to be developed. Alternatively, 4D printing [15,16,17,18] could also offer solutions for minimally invasive implantation procedures, as it uses, for example, shape-memory or smart materials to deliver implants in a folded state before unfolding them to a functional state [41]. In our particular case, however, this is still limited by the need to find a material that not only possess shape-memory properties but is also flexible and has adequate mechanical properties, high biocompatibility, and calcification resistance—the latter two being requirements that are still only met by a limited number of polymers after decades of research since the first polymeric valves in the early 1960s [1,2]. However, this challenge could be overcome by developing suitable polymer blends, which have already been used to introduce or increase shape-memory effects in other materials, like polylactic acid [42,43]. Whether such a composite material that exhibits the combination of necessary properties mentioned above can be found remains a task for polymer scientists for the time being.

## 5. Conclusions

We have succeeded in creating a polymeric heart valve for the aortic position with unidirectional leaflet motion and without the need for restrictors that would obstruct the orifice area. This was achieved by modifying the lateral profile of the triangular leaflets and by 3D printing in silicone. The latest prototype showed performance characteristics comparable to those of the biological and mechanical prostheses currently in use. To build on these promising in vitro results, further research should be conducted to construct similar prototypes from advanced, highly biocompatible and calcification-resistant polymers that may differ in their mechanical properties from the silicone used in this study, before moving on to the first in vitro trials. This could potentially provide a cost-efficient alternative for the replacement of aortic valves.

## Figures and Tables

**Figure 1 polymers-15-04413-f001:**
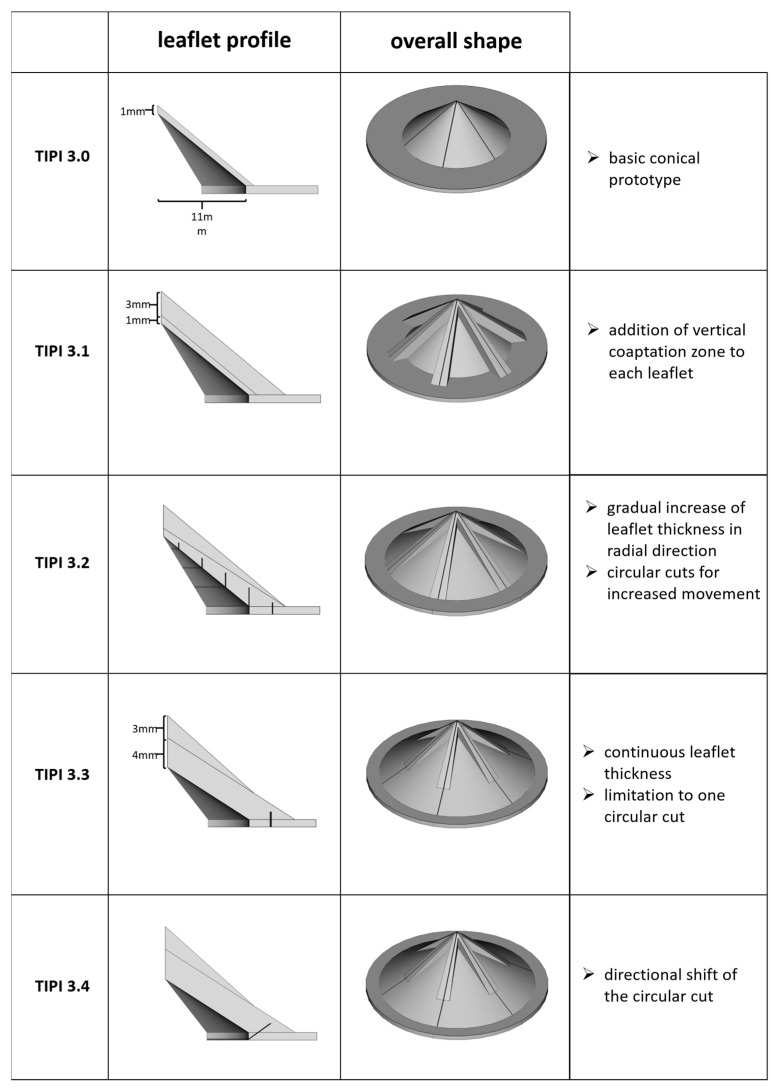
Illustration of the different mentioned prototypes, including the profile of a single leaflet (middle column) as well as the overall prototype shape (right column). The metal restrictor structure included in TIPI 3.0 and 3.1, as presented in our previous work, is not shown.

**Figure 2 polymers-15-04413-f002:**
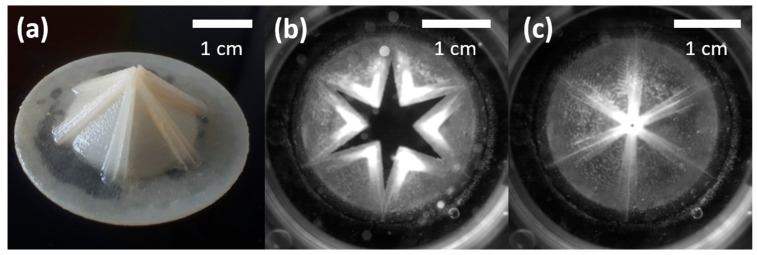
(**a**) Printing result of TIPI 3.4 after removal of the support material. (**b**) Opened and (**c**) closed stages within the hemodynamic testing device. The scale bars refer to each individual picture.

**Figure 3 polymers-15-04413-f003:**
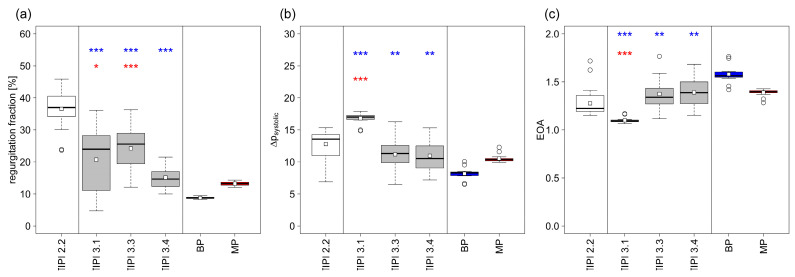
(**a**) Regurgitation fraction, (**b**) mean systolic pressure gradient (Δp_systolic_), and (**c**) effective orifice area (EOA) of newly developed prototypes (TIPI 3.1, 3.3, and 3.4) compared to published results (TIPI 2.2) as well as a bioprosthesis (BP) and a mechanical prosthesis (MP). TIPI 3.0 and 3.2 are not shown due to failure of valve closure. The significance of differences in the performance values compared to BP (blue) and MP (red) is given as follows: * *p* < 0.05, ** *p* < 0.01, *** *p* < 0.001. The significance of differences between the benchmarks (TIPI 2.2, BP, and MP) is not included.

**Table 1 polymers-15-04413-t001:** Performance characteristics of newly developed prototypes (TIPI 3.1, 3.3, and 3.4) compared to published results (TIPI 2.2) as well as a bioprosthesis (BP) and a mechanical prosthesis (MP). TIPI 3.0 and 3.2 are not shown due to failure of valve closure.

	Closing Time (ms)	Closing Volume (mL)	Leakage (mL)	Cardiac Output (L/min)	Regurgitation Fraction (%)	Δp_systolic_ (mmHg)	EOA (cm^2^)
TIPI 2.2	102.3+/−25.25	9.6+/−2.71	15.91+/−2.38	3.1+/−0.25	36.56+/−5.04	12.78+/−2.2	1.28
TIPI 3.1	39.4+/−5.07	3.23+/−0.33	11.23+/−7.57	3.87+/−0.53	20.72+/−10.89	16.82+/−0.86	1.10
TIPI 3.3	89.43+/−21.08	8.67+/−2.65	8.24+/−3.44	3.7+/−0.35	24.24+/−7.2	10.36+/−3.94	1.37
TIPI 3.4	80.53+/−20.48	7.25+/−2.26	3.48+/−1.61	4.22+/−0.34	15.17+/−3.67	10.96+/−2.68	1.39
BP	39.8+/−4.75	2.58+/−0.31	3.54+/−0.32	4.45+/−0.02	8.79+/−0.3	8.18+/−0.9	1.58
MP	42+/−5.32	2.73+/−0.63	6.48+/−0.47	4.23+/−0.03	13.23+/−0.66	10.53+/−0.63	1.38

## Data Availability

The raw data required to reproduce these findings are available to download at https://data.mendeley.com/datasets/kw6m6ztgh3/1 (accessed on 13 July 2023). No processing of these data beyond the calculation of means and standard deviations has been performed. A high-speed video of the presented prototype during a simulated heart cycle within the hemodynamic testing setup can be accessed at https://data.mendeley.com/datasets/6r8f26t6fp/1 (accessed on 17 August 2023).

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
