# Peer review of "Progress on a Novel, 3D-Printable Heart Valve Prosthesis"

_polymers, 2023, doi:10.3390/polym15224413_

Round 1
Reviewer 1 Report (Previous Reviewer 2)
Comments and Suggestions for Authors
The abstract is well edited. But it is suggested to write the abstract according to the format of the journal and in a continuous manner. Also, the abstract can be made more attractive by using more quantitative data. It is suggested to provide more quantitative results.
The manuscript needs general writing and grammar editing. The novelty and purpose of the research should be clearly stated in the abstract and introduction.
Use the following resources to deepen the introduction and discussion. Shape memory performance assessment of FDM 3D printed PLA-TPU composites by Box-Behnken response surface methodology. 4D printing of PLA-TPU blends: effect of PLA concentration, loading mode, and programming temperature on the shape memory effect.
Add scale bar to Figure 2.
It is suggested to summarize the printing parameters in a table. Also, the process of selecting parameters should be presented. Explain how to choose optimum printing parameters.
How is the print quality checked? Explain more about the printing process. How has the reproducibility of these results been checked?
The results section, which is the main part of the article, is completely superficial, summary and short. It is suggested to combine sections 3 and 4 to cover this weakness.
The conclusion is not presented as one of the main parts of the article and is forgotten.
Comments on the Quality of English LanguageNo comment.
Author Response
Response to Reviewer:
Dear Reviewer,
First of all, we want to thank you for your constructive criticism that will help to improve our manuscript.
The abstract is well edited. But it is suggested to write the abstract according to the format of the journal and in a continuous manner. Also, the abstract can be made more attractive by using more quantitative data. It is suggested to provide more quantitative results.
As mentioned in our reply to the first review, the template provided by Polymers recomments a structured paragraph with a format as given in our manuscript. But maybe there is some kind of misunderstanding. What exactly do you mean with “a continuous manner”? We had to shorten the abstract a little bit to stay closer to the recommendation of approximately 200 words given by the author guidelines. We contacted the editor regarding this matter to clear up, if we should remain with the author template or should divert from it to write a continuous abstract as proposed but have not yet received a final answer.
We added the quantitative results regarding the effective orifice area to the abstract.
The manuscript needs general writing and grammar editing.
We edited the whole manuscript. As most changes were small, we did not specifically mark them as changes and only highlighted changes regarding the content.
The novelty and purpose of the research should be clearly stated in the abstract and introduction.
We added statements to line 13-14 as well as 48-50
Use the following resources to deepen the introduction and discussion. Shape memory performance assessment of FDM 3D printed PLA-TPU composites by Box-Behnken response surface methodology. 4D printing of PLA-TPU blends: effect of PLA concentration, loading mode, and programming temperature on the shape memory effect.
We included a paragraph in lines 285-289 considering the two mentioned resources. We don’t currently see the necessity to discuss shape memory effects and FDM printing in more depth at the moment though. We highlighted that the printer used employs an inkjet technology in line 95-96.
We included a paragraph discussing the used test fluid and possible alternatives in lines 261-273.
Add scale bar to Figure 2.
The scale bar was added
It is suggested to summarize the printing parameters in a table. Also, the process of selecting parameters should be presented. Explain how to choose optimum printing parameters.
We included some information in 95-96. The Keyence Agilista uses an inkjet technology therefore a standard setup of printing parameters that we did not change and that is not easily accessible from the documentation of the device. I tried to contact the manufacturer to get more input on the exact parameters but did not yet receive an answer. If more than the stated two parameters (layer thickness and resolution) can be given, we will include a table. But for now, we think a separate table for two values is not feasible.
How is the print quality checked? Explain more about the printing process. How has the reproducibility of these results been checked?
We included a paragraph from line 98 to 105 as well as a half sentence in line 138-139.
The results section, which is the main part of the article, is completely superficial, summary and short. It is suggested to combine sections 3 and 4 to cover this weakness.
We expanded the result section including the most relevant p-values as well as discussing pros and cons of each of the newly developed prototypes in lines 187-200. It is our view though that deeper discussion of these results should remain in the discussion section and we hope this finds your approval.
During revision we noticed a slight error in figure 2b, were the values for TIPI 3.3 were not completely correctly included in the graph and corrected it. We apologize for this mistake.
Several shorter paragraphs and sentences were added to bolster the introduction, result and discussion section and are highlighted as such. The general size of the article was expanded to >4000 words to meet the editors guidelines.
The conclusion is not presented as one of the main parts of the article and is forgotten.
We included a conclusion in lines 290-300.

Reviewer 2 Report (Previous Reviewer 1)
Comments and Suggestions for Authors
This revised version has been improved
Author Response
Dear Reviewer,
thank you for your time and your valuable constructive criticism.
Yours sincerely,
Filip Schröter
This manuscript is a resubmission of an earlier submission. The following is a list of the peer review reports and author responses from that submission.
Round 1
Reviewer 1 Report
Comments and Suggestions for Authors
First of all, congratulations on the improvement of this TIPI valve and on the submission of your paper. However, there are still improvements to be made.
1) Is the Valve going to be used for Aortic?Pulmonary valve ? please explain.
2) The content of the previously PUBLISHED paper needs to be included to make it easier for readers to understand. Please describe in more detail what problems and how they have changed since the first model.
3) You mention the use for patients in developing countries, but you are not at all convincing about how much more costeffective than existing valves and why your valves will be more useful than existing valves.Please explain.
4) There is no explanation of Fig and table in the results area. Please describe all of them.
5) There is not enough discussion derived from the results. It is just a general commentary. Please improve it. 
6)Did you confirm the change in durability of the Valve by changing its design in vitro?
(7) You have increased the height of the VALVE coaptation, but do you think this will not cause problems such as hemolysis? 
Reviewer 2 Report
Comments and Suggestions for Authors
Basic principles in abstract writing are not followed. The background is not included in the abstract. The abstract must be continuous. It is not customary to use references in the abstract. The use of abbreviations is allowed only if they are fully defined in the first reference.
In addition, the abstract is written very poor and incomprehensible. The abstract does not present quantitative data and novelty in a transparent manner.
The introduction is brief, superficial, and incomplete. The number of used and reviewed references is minimal. Also, the paragraphs presented are primarily general and general information. At the end of the introduction, a suitable summary of the importance of the present issue should be provided.
Use the following resources to deepen the introduction and discussion.
4D printing of PET-G via FDM including tailormade excess third shape. Shape memory performance of PETG 4D printed parts under compression in cold, warm, and hot programming. Toughening PVC with Biocompatible PCL Softeners for Supreme Mechanical Properties, Morphology, Shape Memory Effects, and FFF Printability.
The research method is very long and tedious.
In the research method section, the following items should be added. Image of actual printed samples. Providing printing parameters and how to choose them. How to check reproducibility of results and printability.
Add a toolbar to the images in Figure 2. What is the purpose of presenting these images in the results section? It is suggested to be transferred to the research method section.
The results are very superficial and summary. The discussion section has the same problem.
The title of 5th part of the has been forgotten. The conclusion section is not presented either.